

# Designing defensive techniques to handle adversarial attack on deep learning based model

Dhairya Vyas and Viral V. Kapadia

Computer Science and Engineering, The Maharaja Sayajirao University of Baroda, Vadodara, Gujarat, India

## ABSTRACT

Adversarial attacks pose a significant challenge to deep neural networks used in image classification systems. Although deep learning has achieved impressive success in various tasks, it can easily be deceived by adversarial patches created by adding subtle yet deliberate distortions to natural images. These attacks are designed to remain hidden from both human and computer-based classifiers. Considering this, we propose novel model designs that enhance adversarial strength with incorporating feature denoising blocks. Exclusively, proposed model utilizes Gaussian data augmentation (GDA) and spatial smoothing (SS) to denoise the features. These techniques are reasonable and can be mixed in a joint finding context to accomplish superior recognition levels versus adversarial assaults while also balancing other defenses. We tested the proposed approach on the ImageNet and CIFAR-10 datasets using 10-iteration projected gradient descent (PGD), fast gradient sign method (FGSM), and DeepFool attacks. The proposed method achieved an accuracy of 95.62% in under four minutes, which is highly competitive compared to existing approaches. We also conducted a comparative analysis with existing methods.

## INTRODUCTION

Image identification, audio processing, natural language processing, and several other domains have all been significantly impacted by advancements made possible by deep learning algorithms. However, as deep learning has become more widespread, adversarial attacks have also arisen as a substantial danger to the privacy and security of networks. This research is motivated by the critical necessity to fortify deep learning models against adversarial threats to ensure their safety and reliability. This research enhances the defense of deep learning models against adversarial attacks by proposing a novel preventative approach combining adversarial training and feature squeezing. This contribution aims to bolster the resilience of image classification techniques, mitigating the impact of malicious exploitation and advancing the security of practical deep learning implementations. In recent years (*Kumar et al., 2023*; *Li et al., 2023*; *Park, Lee & Kim, 2023*; *Lu et al., 2021*), adversarial attacks on image classification algorithms have developed into a substantial

Corresponding author
Dhairya Vyas,
dhairya.vyas-cse@msubaroda.ac.in

challenge. The safety and dependability of these models have become extremely important issues because of the growing use of deep learning in image processing applications, including medical imaging, face recognition, and object identification. The input pictures may be manipulated by adversarial assaults by introducing undetectable perturbations, which can then cause the models to make predictions that are incorrect or correct. These assaults may have devastating effects, particularly in mission-critical applications like autonomous cars, medical diagnostics, and national security (*Li et al., 2021*). Image classification methods are susceptible to being fooled by adversarial attacks, which may then lead to inaccurate predictions, which in turn might cause accidents or wrong diagnoses. Additionally, hostile instances may be created automatically and readily spread, making it difficult to identify and prevent them from occurring in the first place. In addition, it has been shown that the efficiency of adversarial attacks is consistent across several distinct kinds of picture classification models and architectural frameworks. This indicates that an opponent who is unaware of a trained model's core parameters or architecture may nevertheless be able to launch an assault against the model (*Zanfir & Sminchisescu, 2021*). As a consequence of this, ensuring the safety of deep learning models against attacks from adversaries has become an extremely important challenge, and there is a pressing requirement for the development of efficient preventative methods that will enhance the robustness of these models when faced with adversarial attacks. These assaults have the potential to have significant repercussions, particularly in applications such as driverless cars, medical diagnostics, and the detection of financial fraud (*Han et al., 2021a*).

There are various types of gradient-based attacks, which include Projected Gradient Descent (PGD), Fast Gradient Sign Method (FGSM), and DeepFool, are the most successful and extensively used forms of adversarial attacks. These attacks alter the input data by making use of the gradient information of the target model. This results in an adversarial example that may trick the model into making an incorrect prediction. Researchers have suggested several preventative strategies, such as defensive distillation, adversarial training, and feature squeezing (*Zhang, Gao & Li, 2021*), to defend against the assaults that are being launched by adversaries. However, these methods have several drawbacks, including a high processing cost, decreased accuracy, and susceptibility to various kinds of assault. In this research study, we give a complete examination of the efficiency of several preventative strategies for dealing with gradient-based adversarial assaults. This analysis was carried out by us. We conduct an analysis of the effectiveness of defensive distillation, adversarial training, and feature squeezing on the three gradient-based assaults that are most often used: PGD, FGSM, and DeepFool. In addition, we provide a unique preventative method for improving the resilience of deep learning models against adversarial assaults. This method makes use of a mix of adversarial training and feature squeezing to achieve this goal. Image classification systems may be subjected to adversarial assaults by making minute changes to the images they analyze. This causes the algorithms to provide inaccurate predictions. These changes, which may be disregarded as unimportant "noise" in the image, may in fact be rather effective even against the most sophisticated convolutional network-based techniques (*Xie et al., 2023*; *Li et al., 2023*). This highlights the enormous chasm that exists between the computations carried out by convolutional networks and those carried

out by human brains and poses a significant threat to the security of their practical implementations (*Kumar et al., 2023*). This conclusion provides us with the justification we need to investigate feature denoising techniques as a means of protecting convolutional networks from potentially harmful inputs. By integrating Gaussian data augmentation with SS filters, we provide unique approaches for denoising feature maps (*Lin et al., 2023*; *Lu et al., 2021*).

In summary, the main contributions of this paper lie in the development of effective countermeasures aimed at safeguarding widely used image classification techniques such as AlexNet, VggNet, and ResNet against exploitation by malicious actors. The recommendation to mix denoising algorithms for diverse objects within an image serves to enhance overall model performance. The article's structure includes background information in the second section, an exploration of different categorization models in Section 'Types of dataset and classification models', presentation of the suggested preventative approach and its implementation details in Section 'Novel Defensive Approach', experimental findings and analysis in Section 'Results', and potential avenues for further investigation outlined in Section 'Discussion'.

## RELATED WORK

This literature study provides an overview of recent research in adversarial attacks and defenses for deep learning-based image classification. The study covers a range of defense techniques, including pre-processing techniques, transfer learning, gradient regularization, and generative adversarial networks.

*Xie et al. (2023)* investigate the impact of adversarial examples on image classification models and compare different defense strategies. The limitation of this work is that it does not consider more complex and realistic attack scenarios. *Kumar et al. (2023)* compares different defense techniques for deep learning-based image classification against adversarial attacks. The advantage of this work is that it considers a wide range of attack scenarios and evaluates the effectiveness of defenses against them. *Li et al. (2023)* proposes an adversarial defense method based on pre-processing techniques for image classification. The advantage of this work is that it can effectively defend against adversarial examples generated by different types of attacks, but the limitation is that it may introduce noise to the original image. *Wang et al. (2023)* propose a method for explainable defensive adversarial attacks on image classification. The advantage of this work is that it can help to understand the underlying mechanism of adversarial attacks and improve the robustness of classification models. *Park, Lee & Kim (2023)* proposes a method for detecting and removing adversarial examples in image classification using generative adversarial networks. The advantage of this work is that it can effectively remove adversarial perturbations while maintaining the original image content, but the limitation is that it may introduce artifacts to the image.

*Lin et al. (2023)* proposes an adversarial defense method based on transfer learning for image classification. The advantage of this work is that it can leverage the knowledge learned from pre-trained models to improve the robustness of classification models against adversarial attacks. *Zhang, Chen & Xu (2023)* investigate the limits of defensive adversarial

attacks on image classification models. The advantage of this work is that it provides insights into the limitations of current defense techniques and identifies the challenges for developing more robust defense strategies against adversarial attacks. *Lu et al. (2021)* proposes a defense framework using gradient regularization to enhance the robustness and privacy of deep learning models. The advantage of this structure is that one capability to prevent adversarial attacks by constraining the gradients during training. However, the limitation is that it may decrease the accuracy of the model due to the added constraints. *Su, Chen & Liu (2021)* conducts a survey on defensive adversarial attacks for deep learning-based image classification. The advantage of this survey is that it provides a comprehensive overview of existing defense techniques against adversarial attacks. However, the limitation is that it does not propose any new defense technique. *Xu et al. (2021)* propose a method to improve the robustness of image classification against adversarial attacks using deep convolutional networks with multiple branches. The advantage of this method is that it can improve the accuracy of the model while enhancing its robustness. However, the limitation is that it increases the complexity of the model.

*Hu et al. (2021)* propose a defense technique *via* multi-scale feature enhancement to improve the robustness of deep learning models. The advantage of this technique is its ability to enhance robustness without decreasing the accuracy of the model. However, the limitation is that it may increase the computational cost of the model. *Li et al. (2021)* and *Liu et al. (2021)* reviews adversarial examples detection in deep learning-based image classification. The advantage of this review is its comprehensive analysis of existing detection methods for adversarial examples. However, the limitation is that it does not propose any new detection technique. *Fawzi et al. (2021)* proposed a model-agnostic meta-learning approach to improve adversarial defense. The advantage of this approach is its ability to generalize across different models and tasks. However, it requires many training examples and is computationally expensive. *Wang, Guo & Chen (2021)* proposed a hidden space restriction approach to enhance adversarial defense. The advantage of this approach is its simplicity and effectiveness in defending against a wide range of attacks. However, it may result in a loss of accuracy on some datasets.

*Wu, Chen & Ma (2021)* proposed a disentangled representation approach to improve adversarial defense. The advantage of this approach is its ability to separate robust and non-robust representations, which can improve the model's generalization performance. However, it may require a larger number of training examples. *Zhang et al. (2021)* proposed a defense approach using convolutional block attention modules to enhance the robustness of deep neural networks against adversarial examples. The advantage of this approach is its ability to capture and enhance informative features while suppressing non-informative ones. However, it may result in a reduction in the model's accuracy on certain datasets. *Wang et al. (2021)* and *Zanfir & Sminchisescu (2021)* propose a method for adversarial defense by learning anomaly detectors. The method works by training a deep neural network to detect anomalies in the data, including adversarial examples. Limitations include the need for a large amount of labeled data, which may be expensive or difficult to obtain. Advantages include improved robustness against a wide range of adversarial attacks, and the ability to generalize to different types of anomalies beyond adversarial

examples. *Zhang, Gao & Li (2021)* propose a method for adversarial defense using multi-scale residual learning. The method works by training a deep neural network with residual connections at multiple scales, and then using adversarial training to further improve robustness. Limitations include the need for a larger number of parameters and increased computational complexity, which may be prohibitive in some applications. Advantages include improved robustness against a wide range of adversarial attacks and the ability to generalize to different types of models and architectures. *Han et al. (2021a)* and *Han et al. (2021b)* proposes a method for improving the robustness of convolutional neural networks by using random projections and adversarial training. The method works by projecting the input data onto a lower-dimensional subspace using random projections, and then training the network using adversarial examples generated in the subspace. Limitations include increased computational complexity due to the use of random projections, and the need for careful selection of the projection dimensions. Advantages include improved robustness against a wide range of adversarial attacks and the ability to generalize to different types of models and architectures. *Lal et al. (2021)* aimed to enhance the recognition accuracy of diabetic retinopathy (DR) using adversarial training and feature fusion techniques. DR is a significant cause of blindness worldwide, and early detection is crucial for effective management and treatment. The use of deep learning algorithms in DR diagnosis has shown promising results, but the performance of these algorithms can be impacted by adversarial attacks.

## Types of dataset and classification models
### Datasets
ImageNet is a comprehensive image database that follows the WordNet hierarchy, where each node is depicted by an extensive collection of hundreds of thousands of images. On average, each node contains around 500 images, making it an invaluable resource for researchers, educators, students, and anyone in need of a large image dataset.

The CIFAR-10 dataset is made up of 60,000 color images of size $32 \times 32$ pixels, distributed into 10 different classes, with each class having 6,000 images. There are 50,000 pictures allocated for the train and 10,000 pictures allocated for the test. The dataset is partitioned into six batches, with one batch for testing and the other five for training. Each batch contains 10,000 images, with the test batch containing 1,000 images selected randomly from each class. The images in the training batches are sorted randomly and may have more images from one class than another, but in total, each training batch has an equal distribution of 5,000 images per class.

### AlexNet
According to the AlexNet study, a large perceptron recurrent neural network (RNN) could achieve excellent performance on a challenging dataset by relying solely on supervised learning techniques. This study led to the launch of a competition the year after the introduction of AlexNet, which is still ongoing today. Table 1 shows the AlexNet model architecture.

Every single image in the ImageNet collection was labelled using convolutional neural networks (CNN). Especially with the introduction of AlexNet in 2004, which CNN created

**Table 1  AlexNet model architecture.**

| Type of layer | Size of output | Filter size / stride |
|---|---|---|
| Input | $227 \times 227 \times 3$ | – |
| Convolution | $55 \times 55 \times 96$ | $11 \times 11 / 4$ |
| Max Pooling | $27 \times 27 \times 96$ | $3 \times 3 / 2$ |
| Convolution | $27 \times 27 \times 256$ | $5 \times 5 / 1$ |
| Max Pooling | $13 \times 13 \times 256$ | $3 \times 3 / 2$ |
| Convolution | $13 \times 13 \times 384$ | $3 \times 3 / 1$ |
| Convolution | $13 \times 13 \times 384$ | $3 \times 3 / 1$ |
| Convolution | $13 \times 13 \times 256$ | $3 \times 3 / 1$ |
| Max Pooling | $6 \times 6 \times 256$ | $3 \times 3 / 2$ |
| Flatten | 9,216 | – |
| Dense | 4,096 | – |
| Dense | 4,096 | – |
| Dense | 1,000 | – |
| Output | 1,000 | – |

in partnership with the National Institutes of Health, CNN shook up the world of scientific research. AlexNet can be easily implemented because of the wide variety of deep learning methods already available.

### VggNet

A 224-by-224-pixel RGB picture is used as input for the neural network's cov1 layer. Later, this data is sent into a sequence of convolutional (conv.) layers that use 33 filters to pick up left/right, up/down, and centered spatial information. Sometimes, non-linearity comes first, and then 11 convolution filters are employed to perform linear transformations on the input channels. For 33 conv. layers, the spatial padding is also set to 1 pixel to maintain spatial resolution after convolution, and the convolution stride is also kept at 1 pixel. Over a 22-pixel window with a stride of 2, max-pooling is performed by five layers that follow parts of the conv. layers. Three fully connected (FC) layers follow the convolutional layers; their depths and resulting topologies vary. The first two FC layers have 4,096 channels apiece, while the third performs 1,000-way ILSVRC classification and has 1,000 channels, one for each class. In a neural network, the soft-max layer is the last one. All networks have the same completely linked layers configuration. Table 2 shows the VggNet model architecture.

All hidden layer's bar one includes the rectification (ReLU) non-linearity. Local response normalization (LRN) is not employed in any of the networks since it does not improve performance on the ImageNet dataset, but rather increases memory use and computational time.

### ResNet

There are 50 layers in the deep convolutional neural network known as ResNet-50. There is a pre-trained version of this network that has been taught to recognize objects in the ImageNet database. The ''residual unit'' in ResNet-50 is what makes it stand out; it

**Table 2  VggNet model architecture.**

| Type of layer | Size of output | Size of filter / stride | Filters count |
|---|---|---|---|
| Input | 224 × 224 × 3 | – | – |
| Convolutional | 224 × 224 × 64 | 3 × 3 / 1 | 64 |
| Convolutional | 224 × 224 × 64 | 3 × 3 / 1 | 64 |
| Max Pooling | 112 × 112 × 64 | 2 × 2 / 2 | – |
| Convolutional | 112 × 112 × 128 | 3 × 3 / 1 | 128 |
| Convolutional | 112 × 112 × 128 | 3 × 3 / 1 | 128 |
| Max Pooling | 56 × 56 × 128 | 2 × 2 / 2 | – |
| Convolutional | 56 × 56 × 256 | 3 × 3 / 1 | 256 |
| Convolutional | 56 × 56 × 256 | 3 × 3 / 1 | 256 |
| Convolutional | 56 × 56 × 256 | 3 × 3 / 1 | 256 |
| Max Pooling | 28 × 28 × 256 | 2 × 2 / 2 | – |
| Convolutional | 28 × 28 × 512 | 3 × 3 / 1 | 512 |
| Convolutional | 28 × 28 × 512 | 3 × 3 / 1 | 512 |
| Convolutional | 28 × 28 × 512 | 3 × 3 / 1 | 512 |
| Max Pooling | 14 × 14 × 512 | 2 × 2 / 2 | – |
| Convolutional | 14 × 14 × 512 | 3 × 3 / 1 | 512 |
| Convolutional | 14 × 14 × 512 | 3 × 3 / 1 | 512 |
| Convolutional | 14 × 14 × 512 | 3 × 3 / 1 | 512 |
| Max Pooling | 7 × 7 × 512 | 2 × 2 / 2 | – |
| Fully Connected | 1 × 1 × 4,096 | – | 4,096 |
| Fully Connected | 1 × 1 × 4,096 | – | 4,096 |
| Fully Connected | 1 × 1 × 1,000 | – | 1,000 |
| Output | 1 × 1 × 1,000 | – | – |

connects two layers and serves as a kind of bridge between them. Even if the building is demolished, the data may still travel through unaltered. This design allows the network to ensure minimum harm to the output while reducing muscular endurance to zero in the unavailable slabs at different sizes. Table 3 shows the ResNet model architecture.

There are two recommendations for creating ResNet architecture. Firstly, regardless of the size of the final feature map, the number of filters in each layer remains constant. Second, the time complexity is kept constant by doubling the number of filters if the feature map's size is reduced by half. To limit the number of parameters and matrix multiplications, ResNet-50 employs a bottleneck design for its building blocks, which includes a 1x1 convolution. Consequently, training time for each layer may be reduced. ResNet-50 is a model that employs a stack of three layers in each block, as opposed to the two layers used by other ResNet models.

### Novel defensive approach
The novel strategy makes use of a feature-map that is split into two shares. The primary function of this system is to locate and label all object mappings inside the bounds of the input image, as shown in the Figure. The final detection result for the input picture is generated by the second part, which combines all feature-map portions, as shown in Fig. 1.

**Table 3  ResNet model architecture.**

| Type of layer | Shape of output | No. of parameters |
|---|---|---|
| Input | 224 × 224 × 3 | 0 |
| Conv1 | 112 × 112 × 64 | 9,408 |
| BatchNorm1 | 112 × 112 × 64 | 256 |
| Activation1 | 112 × 112 × 64 | 0 |
| MaxPooling | 56 × 56 × 64 | 0 |
| Res2a_Branch2a | 56 × 56 × 64 | 4,096 |
| Res2a_Branch2b | 56 × 56 × 64 | 36,864 |
| Res2a_Branch2c | 56 × 56 × 256 | 16,384 |
| Res2a_Branch1 | 56 × 56 × 256 | 16,384 |
| Res2a | 56 × 56 × 256 | 0 |
| Res2b_Branch2a | 56 × 56 × 256 | 16,384 |
| Res2b_Branch2b | 56 × 56 × 256 | 36,864 |
| Res2b_Branch2c | 56 × 56 × 256 | 16,384 |
| Res2b | 56 × 56 × 256 | 0 |
| Res2c_Branch2a | 56 × 56 × 256 | 16,384 |
| Res2c_Branch2b | 56 × 56 × 256 | 36,864 |
| Res2c_Branch2c | 56 × 56 × 256 | 16,384 |
| Res2c | 56 × 56 × 256 | 0 |
| Res3a_Branch2a | 28 × 28 × 256 | 65,536 |
| Res3a_Branch2b | 28 × 28 × 128 | 32,768 |
| Res3a_Branch2c | 28 × 28 × 512 | 66,048 |
| Res3a_Branch1 | 28 × 28 × 512 | 131,072 |
| Res3a | 28 × 28 × 512 | 0 |
| Res3b_Branch2a | 28 × 28 × 512 | 262,144 |
| Res3b_Branch2b | 28 × 28 × 128 | 65,536 |
| Res3b_Branch2c | 28 × 28 × 512 | 262,144 |
| Res3b | 28 × 28 × 512 | 0 |
| Res3c_Branch2a | 28 × 28 × 512 | 262,144 |
| Res3c_Branch2b | 28 × 28 × 128 | 65,536 |
| Res3c_Branch2c | 28 × 28 × 512 | 262,144 |
| Res3c | 28 × 28 × 512 | 0 |
| Res3d_Branch2a | 28 × 28 × 512 | 262,144 |
| Res3d_Branch2b | 28 × 28 × 128 | 65,536 |
| Res3d_Branch2c | 28 × 28 × 512 | 262,144 |
| Res3d | 28 × 28 × 512 | 0 |
| Res4a_B | – | – |

## Algorithm

**Step 1:** Receive the input image for analysis.

**Step 2:** Split the feature-map into two halves.

Utilize the first part to locate and label object mappings within the input image.

**Step 3:** Combine both feature-map portions to generate the final detection result for the input image.

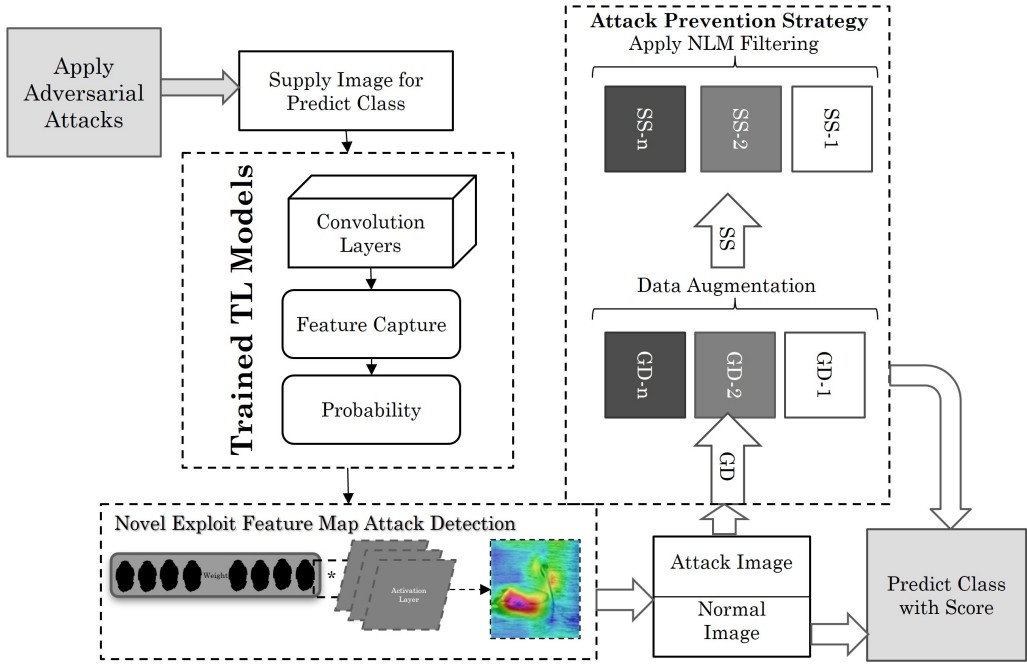

**Figure 1** **Novel defensive approach.** Image source credit: Pelican, CC0 1.0, Peakpx.

**Step 4:** Identify input variables (x) for the black box function f(x).

**Step 5:** Use saliency to pinpoint regions of the input image contributing to the result of the black box function. Remove regions R of the image and observe the response of the function for local explanations.

**Step 6:** Assign a scalar value to each pixel using a mask function. Create a perturbation operator $''\Phi(X_0; m)(u)$ to introduce perturbations by replacing regions with constants, adding noise, or blurring the image.

**Step 7:** Implement Gaussian Data Augmentation (GDA) by adding Gaussian noise to original samples. Expand the dataset to enhance model effectiveness and visualize adversarial attacks.

**Step 8:** Implement Spatial Smoothing (SS) using Non-Local Means (NLM) for each color channel independently. Utilize a median-filtered form for components to smooth the image and eliminate threatening signals.

**Step 9:** Combine adversarial training and feature squeezing to improve the resilience of deep learning models against adversarial attacks.

**Step 10:** Obtain the final output, representing the protected and enhanced deep learning model against adversarial threats.

### Description

First determining the input variables of x that will be utilized to investigate a black box function f(x) is a prerequisite to developing an explanatory rule for $f(x)$. Saliency is used to pinpoint regions of an input picture $x_0$ that contribute to the result f generated by the

black box ($x_0$). This may be done by removing regions R of $x_0$ and monitoring how f($x$) responds. For instance, if the picture is represented by f($x_0$) = +1, then attack is deleted owing to the selection of R. As such, we anticipate that this explanation will characterize the connection between f and $x_0$, and hence recognize it as a local explanation.

The idea may seem straightforward, but there are really some complications with it. Defining what it means to "delete" information is one of the primary issues. We want to emulate natural or believable imaging effects to produce more meaningful perturbations and explanations. Since we can't alter the picture creation process, we resort to three tried-and-true methods: replacing region R with a constant number, adding noise, and blurring the image.

By assigning a scalar value m to each pixel u in using a mask function m (0,1), we may formally create a perturbation operator ($u$).

$$\Phi(X_0; m)(u) = \begin{cases} m(u)x_0(u) + (1-m(u))\mu_0, \text{Constant} \\ m(u)x_0(u) + (1-m(u))n(u), \text{Noise} \\ \int g_{\sigma_0}m(u)(v-u)x_0(v)\,dv, \text{Blur}. \end{cases} \tag{1}$$

The suggested approach uses the values 0 for the mean color, $n$ ($u$) for the number of independently and identically distributed (i.i.d.) Gaussian noise samples per pixel, and 0 for the maximum isotropic standard deviation of the Gaussian blur kernel $g$. Note that picking 0 ->10 produces a very blurry picture. The benefit of this method is that the ensuing visualizations make use of adversarial assaults in a very transparent way. This is a huge plus as a minimal mask is generated when trying to hide an object from the network's recognition.

In computer vision, GDA is a popular method for making models more resistant to manipulation by malicious users. Adding Gaussian noise to a copy of the original samples is one way to expand the dataset. As a bonus, it may be used to introduce Gaussian noise into a sample without the need for additional data. This method is useful since it may be used to any kind of assault. Its primary function, however, is to supplement the training set to boost the effectiveness of the model.

Whereas SS is an image-centric defense. By using non-local means (NLM) spatial smoothing, it seeks to eliminate threatening signals. Denote the parts of $x$ as, $xijk$. Keep in mind that $i$ the width index, $j$ the height index, and $k$ the color channel. For each given $w$, the median-filtered form of the component, $x_{ijk}$. is used instead:

$$x_{ijk} \leftarrow \text{median}\{x_{ijk} : i - \lceil w/2 \rceil \leq i' \leq i + \lceil w/2 \rceil - 1, j - \lceil w/2 \rceil \leq j' \leq j + \lceil w/2 \rceil - 1\} \tag{2}$$

$$x_{ijk} = x_{(1-i),j,k} \qquad \text{for } i = 0, -1, \tag{3}$$

$$x_{(k1+i),j,k} = x_{(k1+1-i),j,k} \qquad \text{for } i = 1, 2, \tag{4}$$

Where border characteristics are mirrored when necessary and so for $j$. Please consider that the local spatial smoothing is implemented independently for each $k$-color channel. At last

Vyas and Kapadia (2024), *PeerJ Comput. Sci.*, DOI 10.7717/peerj-cs.1868

Complexity of model can be calculated with MACs and FLOPs. Counting flops (skipping activations for now, anyways only use RELUs):

Linear layers:

$$MAC(multiply - accumulate) = output.shape * input.shape \tag{5}$$

$$ADD = output.shape(for\ bias) \tag{6}$$

$$Flops = 2 * MAC + ADD \tag{7}$$

Convolutional layers:

$$N\ conv\ OPS = input_h * input_w / stride = output^2 \tag{8}$$

$$MAC/filter = kernel\ size\ 2 * Input\ Channels * Output\ Channels \tag{9}$$

$$ADD = Output\ Channels\ (for\ bias) \tag{10}$$

$$Flops = 2 * (MAC/filter * N\ conv\ OPS) + ADD. \tag{11}$$

## RESULTS

The experiment results utilizing the algorithm was conducted on Google Colab, running on an LTS with an Intel(R) Xeon(R) CPU with a processor speed of 2.20 GHz and 13GB of RAM. The CPU has a speed of 2,200.000 MHz, and the cache size is approximately 56,320 KB, while the HDD has a capacity of approximately 33GB. Additionally, a GPU with a capacity of approximately 64GB was utilized. Furthermore, this experiment was taken further for experimental purposes by testing it on a supercomputer of the Maharaja Sayajirao University of Baroda, Baroda which was obtained from the government of Gujarat. This supercomputer is equipped with 96GB of RAM, 16TB of ROM, and an Intel® Xeon® Gold 6145 processor, as well as a 16GB NVIDIA QUADRO RTX 5000 graphics card. The epoch size was increased during this experiment to assess the accuracy of the generated output. The comparative study yielded the following parameters:

**Accuracy:** Accuracy is determined by comparing the model's predictions to the training set's true values. Model accuracy is expressed as a percentage and is calculated by dividing the number of accurate predictions by the total number of predictions.

$$ACC = \frac{Predicted\ Value}{Actual\ Value} * 100\%. \tag{12}$$

**Error rate:** This is a ratio that is derived by comparing the number of properly labelled photos to the number of mistakes.

$$\partial = \left| \frac{V_A - V_E}{V_E} \right| *100\%. \tag{13}$$

where $V_A$ is the observed value, $V_E$ is the anticipated error, and is the percentage error.

**Time:** Maximum time in milliseconds that a single task may use the CPU. It determines how long the task can run at full CPU capacity. It is a nebulous variable that may be set independently for each task or process step.

**Epoch:** The Epoch parameter controls the number of times a dataset is processed during training. It is an important hyperparameter that affects the performance of a model. Setting it too low leads to underfitting, while setting it too high leads to overfitting. The optimal Epoch value depends on various factors and requires experimentation to determine.

Now first we will detect the adversarial attacks using existing methods as well as our proposed novel exploit feature-map it can be shows that our method gives best visualization for attack image. Then second, we will apply different defensive approaches on attack image and check model performance.

## DISCUSSION

This section provides discussion on results. As shown in Table 4 different adversarial attack detection methods calculate original image and attack image feature map. The Novel Exploit Feature-Map (*Saeed et al., 2022*) obtained the elaborate attack image simply. So, from here attack image will be future need to go for removal of attacks in next section using Gaussian data augmentation and sequential smoothing.

As from Tables 5–10 it can be said that Novel Defensive Approach defensive approach converts a pretrained classifier into a robust one, which is effective and straightforward to implement. This approach works for both white-box and black-box settings, and we have validated it through experiments on ImageNet and CIFAR-10. Tables 5–10 results demonstrate that we can convert pretrained models like AlexNet, VggNet, and ResNet on CIFAR-10 and ImageNet into provably robust models. The accuracy of converted models is summarized in Tables 5–10. GDA combined with SS method can increase the accuracy of a pretrained AlexNet on CIFAR-10 to 5%, VggNet to 3%, and ResNet to 3% on ImageNet under adversarial perturbations with 2-norm less than 132/255. Above all experiments conducted on a university's supercomputer to demonstrate the effectiveness of GDA combined with SS method to reduce time complexity.

Table 11 compares different defense models used in deep learning, including Defense AlexNet, Defense VGG19, Defense ResNet50, and Defense DensNet169. Each model is evaluated based on its parameters, Multiply-Accumulate operations (MACs), and Floating Point Operations (FLOPs). The parameters represent the number of learnable weights and biases in the models, while MACs and FLOPs measure the computational complexity involved in processing data during inference. The table provides a concise overview of these metrics, enabling a quick comparison between the different defense models.

Table 4  Adversarial attack detection with novel exploit feature-map.

| Method | Original Image | FGSM Attack | PGD Attack | DeepFool Attack |
|---|---|---|---|---|
| Images Type | | | | |
| Guided Propagation Model Han B et al. (2021) | | | | |
| Smooth Grad Hu Y et al. (2021) | | | | |
| Guided-CAM Mapping Han J et al. (2021) | | | | |
| Inverted Image Representation Lin X et al. (2023) | | | | |
| Novel Exploit Feature-Map Saeed AA et al. (2023) | | | | |

## CONCLUSIONS

The research paper proposes a formal framework for detecting and preventing adversarial deep learning attacks. Adversarial attacks aim to manipulate machine learning models by exploiting their vulnerabilities. These attacks can result in misclassifications, which can be dangerous in applications such as autonomous vehicles and medical diagnosis.

The novel framework utilizes a novel exploit feature-map that can detect adversarial attacks with moderate temporal complexity. The feature-map takes less than 8–9 min to execute and demonstrates superior performance compared to existing methods. In the experiments, the framework achieved an error rate of only 28.42% on a test sample set, while current techniques have error rates exceeding 40%. The novel framework employs a

**Table 5   AlexNet- ImageNet defense on adversarial attack.**

| | | AlexNet-ImageNet | | | | | | | | |
|---|---|---|---|---|---|---|---|---|---|---|
| NO | Preventive | PGD | | | FGSM | | | DeepFool | | |
| | | Accuracy (%) | Error (%) | Time (Sec) | Accuracy (%) | Error (%) | Time (Sec) | Accuracy (%) | Error (%) | Time (Sec) |
| 1 | Spatial Smoothing (SS) | 90.12% | 9.88% | 219 | 94.46% | 5.54% | 190 | 89.21% | 10.79% | 365 |
| 2 | Gaussian Data Augmentation (GDA) | 91.35% | 8.65% | 289 | 92.32% | 7.68% | 276 | 88.37% | 11.63% | 396 |
| 3 | Feature Squeezing (FS) | 89.95% | 10.05% | 276 | 93.19% | 6.81% | 258 | 87.64% | 12.36% | 423 |
| 4 | FS+GDA | 89.23% | 10.77% | 312 | 91.20% | 8.80% | 312 | 88.45% | 11.55% | 412 |
| 5 | Novel Defensive Approach | 94.46% | 5.54% | 224 | 96.23% | 3.77% | 186 | 89.43% | 10.57% | 386 |

**Table 6   AlexNet- CIFAR-10 defense on adversarial attacks.**

| | | AlexNet-CIFAR-10 | | | | | | | | |
|---|---|---|---|---|---|---|---|---|---|---|
| NO | Preventive | PGD | | | FGSM | | | DeepFool | | |
| | | Accuracy (%) | Error (%) | Time (Sec) | Accuracy (%) | Error (%) | Time (Sec) | Accuracy (%) | Error (%) | Time (Sec) |
| 1 | Spatial Smoothing (SS) | 91.23% | 8.77% | 217 | 92.13% | 7.87% | 210 | 88.13% | 11.87% | 383 |
| 2 | Gaussian Data Augmentation (GDA) | 92.28% | 7.72% | 292 | 91.24% | 8.76% | 278 | 88.18% | 11.82% | 412 |
| 3 | Feature Squeezing (FS) | 88.82% | 11.18% | 288 | 92.34% | 7.66% | 267 | 87.22% | 12.78% | 312 |
| 4 | FS+GDA | 88.78% | 11.22% | 322 | 92.42% | 7.58% | 310 | 88.45% | 11.55% | 486 |
| 5 | Novel Defensive Approach | 95.62% | 4.38% | 212 | 97.39% | 2.61% | 196 | 89.12% | 10.88% | 476 |

**Table 7   VggNet- ImageNet defense on adversarial attacks.**

| | | VggNet-ImageNet | | | | | | | | |
|---|---|---|---|---|---|---|---|---|---|---|
| NO | Preventive | PGD | | | FGSM | | | DeepFool | | |
| | | Accuracy (%) | Error (%) | Time (Sec) | Accuracy (%) | Error (%) | Time (Sec) | Accuracy (%) | Error (%) | Time (Sec) |
| 1 | Spatial Smoothing (SS) | 88.23% | 11.77% | 185 | 89.13% | 10.87% | 189 | 86.93% | 13.07% | 371 |
| 2 | Gaussian Data Augmentation (GDA) | 90.28% | 9.72% | 260 | 80.24% | 19.76% | 243 | 86.98% | 13.02% | 380 |
| 3 | Feature Squeezing (FS) | 86.82% | 13.18% | 288 | 90.34% | 9.66% | 225 | 86.02% | 13.98% | 300 |
| 4 | FS+GDA | 88.46% | 11.54% | 299 | 87.10% | 12.90% | 287 | 87.25% | 12.75% | 484 |
| 5 | Novel Defensive Approach | 92.62% | 7.38% | 202 | 96.16% | 3.84% | 191 | 87.92% | 12.08% | 464 |

two-step process for detecting adversarial attacks. First, the input image is passed through the deep learning model to obtain the output classification. Then, the image is passed through the exploit feature-map to identify any adversarial attacks.

The novel exploit feature-map is based on the principle of high-dimensional data clustering. It operates by transforming the input image into a feature space that is highly discriminative for the target classification. Adversarial attacks introduce noise in the image, which can be detected by the feature-map as outlying points. The authors demonstrate

**Table 8  VggNet- CIFAR-10 defense on adversarial attacks.**

| NO | Preventive | VggNet-CIFAR-10 | | | | | | | | |
| | | PGD | | | FGSM | | | DeepFool | | |
| | | Accuracy (%) | Error (%) | Time (Sec) | Accuracy (%) | Error (%) | Time (Sec) | Accuracy (%) | Error (%) | Time (Sec) |
| 1 | Spatial Smoothing (SS) | 90.23% | 9.77% | 194 | 90.13% | 9.87% | 188 | 86.03% | 13.97% | 373 |
| 2 | Gaussian Data Augmentation (GDA) | 91.28% | 8.72% | 268 | 67.24% | 32.76% | 248 | 85.88% | 14.12% | 381 |
| 3 | Feature Squeezing (FS) | 87.82% | 12.18% | 262 | 90.34% | 9.66% | 235 | 84.72% | 15.28% | 294 |
| 4 | FS+GDA | 88.47% | 11.53% | 297 | 87.92% | 12.08% | 286 | 85.85% | 14.15% | 468 |
| 5 | Novel Defensive Approach | 93.62% | 6.38% | 200 | 95.09% | 4.91% | 189 | 86.42% | 13.58% | 461 |

**Table 9  ResNet- ImageNet defence on adversarial attacks.**

| NO | Preventive | ResNet-ImageNet | | | | | | | | |
| | | PGD | | | FGSM | | | DeepFool | | |
| | | Accuracy (%) | Error (%) | Time (Sec) | Accuracy (%) | Error (%) | Time (Sec) | Accuracy (%) | Error (%) | Time (Sec) |
| 1 | Spatial Smoothing (SS) | 92.23% | 7.77% | 240 | 94.13% | 5.87% | 232 | 90.23% | 9.77% | 393 |
| 2 | Gaussian Data Augmentation (GDA) | 93.28% | 6.72% | 316 | 93.64% | 6.36% | 308 | 90.48% | 9.52% | 443 |
| 3 | Feature Squeezing (FS) | 89.82% | 10.18% | 314 | 94.34% | 5.66% | 299 | 89.72% | 10.28% | 330 |
| 4 | FS+GDA | 89.09% | 10.91% | 347 | 96.92% | 3.08% | 334 | 91.05% | 8.95% | 504 |
| 5 | Novel Defensive Approach | 93.62% | 6.38% | 224 | 99.69% | 0.31% | 189 | 91.82% | 8.18% | 491 |

**Table 10  ResNet - CIFAR-10 defense on adversarial attacks.**

| NO | Preventive | ResNet -CIFAR-10 | | | | | | | | |
| | | PGD | | | FGSM | | | DeepFool | | |
| | | Accuracy (%) | Error (%) | Time (Sec) | Accuracy (%) | Error (%) | Time (Sec) | Accuracy (%) | Error (%) | Time (Sec) |
| 1 | Spatial Smoothing (SS) | 94.66% | 5.34% | 238 | 95.01% | 4.99% | 230 | 90.32% | 9.68% | 403 |
| 2 | Gaussian Data Augmentation (GDA) | 92.62% | 7.38% | 313 | 93.72% | 6.28% | 298 | 90.56% | 9.44% | 432 |
| 3 | Feature Squeezing (FS) | 90.27% | 9.73% | 309 | 95.24% | 4.76% | 287 | 89.78% | 10.22% | 332 |
| 4 | FS+GDA | 89.10% | 10.90% | 343 | 96.93% | 3.07% | 330 | 91.14% | 8.86% | 506 |
| 5 | Novel Defensive Approach | 92.94% | 7.06% | 233 | 99.78% | 0.22% | 176 | 91.89% | 8.11% | 454 |

that this approach is effective in detecting a variety of adversarial attacks, including FGSM, PGD, and DeepFool.

To further enhance the framework's performance, the authors propose a filtering technique that combines GDA and SS. GDA is a method that removes outliers by fitting a Gaussian distribution to the data and removing samples that fall outside a certain threshold. SS is a method that smooths the image by applying a sliding window and replacing the center pixel with the average of the surrounding pixels. The authors demonstrate that the

**Table 11 Model complexity in terms of MACs and Flops.**

| Model | Parameters | MACs | FLOPs |
|---|---|---|---|
| Defense AlexNet | 61.101M | 714206912.0 | 0.714G |
| Defense VGG19 | 25.557M | 4133742592.0 | 4.145G |
| Defense ResNet50 | 0.144G | 19632112640.0 | 19.632G |
| Defense DensNet169 | 14.149M | 3436117120.0 | 3.455G |

combined approach is effective in removing noise from adversarial attacks, resulting in improved classification accuracy.

The novel framework is evaluated across multiple deep learning models, including AlexNet, VggNet, and ResNet. The authors demonstrate that the framework achieves an average accuracy of 95.62% across these models and takes less than 4 min to execute. Overall, the proposed framework presents a promising approach for detecting and preventing adversarial attacks on deep learning models. The exploit feature-map and filtering techniques demonstrate superior performance compared to existing methods and can be easily incorporated into existing deep learning pipelines. Further research is needed to evaluate the framework's performance on larger datasets and more complex attacks.

The proposed work has a limitation associated with its reliance solely on pre-trained transfer learning models. While transfer learning is a powerful approach leveraging knowledge from pre-existing models, this constraint may pose challenges in scenarios where the specificities of the target task demand a more tailored or task-specific model. The exclusive use of pre-trained models might limit the adaptability and fine-tuning capabilities, potentially affecting the model's performance in certain specialized domains or datasets. Addressing this limitation could involve exploring hybrid approaches that integrate both pre-trained models and task-specific learning to strike a balance between leveraging existing knowledge and tailoring the model for optimal performance in specific contexts.

## ACKNOWLEDGEMENTS

We would like to extend our heartfelt appreciation to Dr. Apurva Shah, Head of the Computer Science and Engineering Department, MSU, Vadodara, Gujarat for his encouragement and support. His vision and leadership have provided us with a conducive research environment, enabling us to pursue our research goals with dedication. We are thankful to our colleagues and fellow researchers for their constructive discussions and input that have contributed to the refinement of our ideas. The assistance provided by the technical staff and resources of The Maharaja Sayajirao University of Baroda has been instrumental in the successful completion of this research. Lastly, we extend our gratitude to our friends and family for their unwavering encouragement and understanding during this endeavor.

### Funding
The authors received no funding for this work.

### Competing Interests
The authors declare there are no competing interests.

### Author Contributions
- Dhairya Vyas conceived and designed the experiments, performed the experiments, performed the computation work, prepared figures and/or tables, authored or reviewed drafts of the article, and approved the final draft.
- Viral V. Kapadia analyzed the data, performed the computation work, prepared figures and/or tables, authored or reviewed drafts of the article, and approved the final draft.

### Data Availability
The attack image map and attack defense code are available in the Supplemental Files. The imagenet dataset is available at: Available at https://www.image-net.org/.

### Supplemental Information
Supplemental information for this article can be found online at http://dx.doi.org/10.7717/peerj-cs.1868#supplemental-information.

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
