# Peer review of "Designing defensive techniques to handle adversarial attack on deep learning based model"

_PeerJ Computer Science, doi:10.7717/peerj-cs.1868_

## Round 0.1 · original submission · Major Revisions

Please revise the paper according to the reviewer's comments.
**Language Note:** PeerJ staff have identified that the English language needs to be improved. When you prepare your next revision, please either (i) have a colleague who is proficient in English and familiar with the subject matter review your manuscript, or (ii) contact a professional editing service to review your manuscript. PeerJ can provide language editing services - you can contact us at copyediting@peerj.com for pricing (be sure to provide your manuscript number and title). – PeerJ Staff

Reviewer 1 ·

Basic reporting

Please see additional comments.

Experimental design

Please see additional comments.

Validity of the findings

Please see additional comments.

Additional comments

This paper deals with an exciting topic. The article has been read carefully, and some minor issues have been highlighted in order to be considered by the author(s).
#1 What is the motivation of this paper?
#2 What is the contribution and novelty of this paper?
#3 What is the advantage of this paper?
#4 Which evaluation metrics did you used for comparison?
#5 Some paper would be reflected in the related work such as “https://ieeexplore.ieee.org/abstract/document/9580824”, “https://www.hindawi.com/journals/js/2022/4390413/”, “https://search.ieice.org/bin/summary.php?id=e105-d_1_170”, “https://www.hindawi.com/journals/js/2021/6473833/”, “https://www.sciencedirect.com/science/article/pii/S0925231222008219”, “https://ieeexplore.ieee.org/abstract/document/9579036”, “https://link.springer.com/article/10.1007/s11042-022-12941-w”, “https://ieeexplore.ieee.org/abstract/document/10046665”, “https://www.sciencedirect.com/science/article/pii/S0167404822004539”, “https://link.springer.com/article/10.1007/s10489-022-03313-w”.

#6 Meaning of the symbols used can be explained clearly.
#7 The limitation of the proposed work can be discussed.

·

Basic reporting

This research paper is fully interesting and very well described. However, certain aspects of the paper's technicality and presentation prevent me from accepting the paper, and I am marking it as "Major revisions requested". Please consider the following remarks to improve your article.

Experimental design

The discussions should highlight why the proposed method is providing good results

Validity of the findings

The discussions should highlight why the proposed method is providing good results

Additional comments

General Comments
This research paper is fully interesting and very well described. However, certain aspects of the paper's technicality and presentation prevent me from accepting the paper, and I am marking it as "Major revisions requested". Please consider the following remarks to improve your article.

1- The introduction section needs to highlight the motivation contribution of the research.
2- Write general algorithm of the proposed work in steps format so that other can replicate the study.
3- Please separate the result and discussion session. The discussions should highlight why the proposed method is providing good results.
4- I would like authors to present the limitations of the current methods (if any) and possible directions for future research.
5- The discussions should highlight why the proposed method is providing good results.

Reviewer 3 ·

Basic reporting

1.The motivation of introducing the adversarial strength enhance model should be presented in more detail.
2.The authors are suggested to briefly summarize the main contributions of this paper at the end of Introduction.
3.When writing a paper, the author needs to pay more attention to the grammar rules, and please correct the grammatical errors in this manuscript.
4.It will be helpful to refer to the following papers.
Yang J, Zhang Z, Xiao S, et al. Efficient data-driven behavior identification based on vision transformers for human activity understanding[J]. Neurocomputing, 2023, 530: 104-115.
He J, Wen J, Xiao S, et al. Multi-AUV Inspection for Process Monitoring of Underwater Oil Transportation[J]. IEEE/CAA Journal of Automatica Sinica, 2023, 10(3): 828-830.
Yang J, Cheng C, Xiao S, et al. High Fidelity Face-Swapping With Style ConvTransformer and Latent Space Selection[J]. IEEE Transactions on Multimedia, 2023.

Experimental design

1.The authors should add at least two existing datasets to the experiment.
2.The authors should add at least two new classification networks for experiments, not just in AlexNet, VggNet, ResNet.
3.The authors need to add some visual experimental results in this manuscript.

Validity of the findings

In this paper, the authors propose a model designs that enhance adversarial strength with incorporating feature denoising blocks. And they tested the proposed approach on the ImageNet and CIFAR-10 datasets. The result shows a high accuracy. However, some shortcomings in this manuscript.

---

## Round 0.2 · accepted · Accept

According to the comments of reviewers, after comprehensive consideration, it is decided to accept it.

Reviewer 1 ·

Basic reporting

None

Experimental design

None

Validity of the findings

None

Additional comments

I recommend the acceptance.

·

Basic reporting

This research paper is fully interesting and very well described

Experimental design

good

Validity of the findings

Excellent

Reviewer 3 ·

Basic reporting

The revised manuscript has addressed all my concerns and can be accepted.

Experimental design

No more comments.

Validity of the findings

No more comments.